# Volatile Profiles and Sensory Characteristics of Cabernet Sauvignon Dry Red Wines in the Sub-Regions of the Eastern Foothills of Ningxia Helan Mountain in China

**DOI:** 10.3390/molecules27248817

**Published:** 2022-12-12

**Authors:** Xixian Song, Mengqi Ling, Demei Li, Baoqing Zhu, Ying Shi, Changqing Duan, Yibin Lan

**Affiliations:** 1Center for Viticulture and Enology, College of Food Science and Nutritional Engineering, China Agricultural University, Beijing 100083, China; 2Key Laboratory of Viticulture and Enology, Ministry of Agriculture and Rural Affairs, Beijing 100083, China; 3College of Food Science and Nutritional Engineering, Beijing University of Agriculture, Beijing 102206, China; 4College of Biological Sciences and Technology, Beijing Forestry University, Beijing 100083, China

**Keywords:** terroir, sub-regions, Cabernet Sauvignon, volatile compounds, CATA, QDA

## Abstract

To elucidate the effects of the different terroir on wine aroma in six sub-regions of Eastern Foothills of Helan Mountain in Ningxia, a premium wine-producing region in China, 71 Cabernet Sauvignon wines were investigated by gas chromatography-mass spectrometry (GC-MS), check-all-that-apply (CATA), and quantitative descriptive analysis (QDA). The bidirectional orthogonal partial least squares-discriminant analysis (O2PLS-DA) results showed that the Cabernet Sauvignon dry red wines from Xixia (XX) and Yongning (YN) had similar volatile profiles due to their geographical proximity and were characterized by higher concentrations of esters, higher alcohols, and volatile phenols because the similar aromatic profiles were detected in their dry red wines. Shizuishan (SZS) and Hongsipu (HSP) wines showed clear differences compared to the wines of the other four sub-regions, being mainly characterized by relatively higher phenolic aldehydes and volatile phenols. The concentrations of methoxypyrazines and norisoprenoids varied mainly depending on the climate diversity of the sub-regions. The highest 3-isobutyl-2-methoxypyrazine (IBMP) concentration was presented in the Helan (HL) wines. The Qingtongxia (QTX) wines have the highest *β*-damascenone, which might be influenced by the fact that QTX has the lowest effective accumulated temperature and the highest sunshine duration among the five sub-regions. Esters including ethyl octanoate, ethyl decanoate, ethyl butanoate, ethyl hexanoate, and isoamyl acetate were the highest in HL. Additionally, the herbaceous, black berry, and red berry notes in HL and QTX were the most outstanding.

## 1. Introduction

Terroir is the summary of the regional climate, geology, management measures, and soil in a vineyard, which is regarded as the main basis for representing regional wine quality [1,2]. Wines that were selected from different regions with different terroirs can be characterized and distinguished by their sensory characteristics and volatile compounds [3,4,5,6]. For example, Cabernet Sauvignon and Malbec wines from Australia, California, and Argentina were clearly separated by volatile compounds and sensory data through multivariate statistical analysis [7]. Besides, the Cabernet Sauvignon wines from different sub-regions in Australia exhibited diverse sensory characteristics which differed from Bordeaux wines [8]. In China, previous studies focused more on the effect of terroir on wine aroma in the main production areas, such as the Jiaodong Peninsula, Bohai Bay, Loess Plateau, Huaizhuo Basin, Qingxu, Yunnan Plateau, Gansu, Xinjiang, and Ningxia regions [9,10,11,12]. However, the research on the influence of the meso-climates of sub-regions in a specific region on the volatile profiles and sensory characteristics of their wines is limited.

Ningxia Province is located in the interior of northwest China and its terrain is shown in Figure 1. Because of Helan Mountain, the terrain in Ningxia is high altitude in the south and low altitude in the north and descends in a stepped manner [13]. In recent years, the Eastern Foothills of Helan Mountain in Ningxia, a new emerging wine-producing region, has become one of the most important wine-producing regions in China, which represents tremendous development potential. This region is located in front of the Eastern Foothills, between the alluvial fan and the alluvial plain of the Yellow River. With Helan Mountain in the west as the natural barrier against the cold current from the Siberian and the Yellow River irrigation channel crossing in the east, the environmental conditions and soil characteristics of the wineries vary greatly depending on their proximity to the mountain [14]. In addition, the Eastern Foothills of Helan Mountain region belongs to the mid-temperate semi-arid climate zone, which has sufficient sunshine, abundant heat, a high diurnal temperature range, and convenient irrigation conditions [13]. It can be divided into six sub-regions, including Shizuishan (SZS), Helan (HL), Xixia (XX), Yongning (YN), Qingtongxia (QTX), and Hongsipu (HSP), which are adjacent to each other in order from north to south. However, there are some sub-regions with only a few wineries at the moment, such as the SZS sub-region, where only two wineries have been established. Cabernet Sauvignon, the predominant grape variety in Ningxia, is used to make dry red wines with complex aromas and diverse characteristics [14]. Recently, researchers have begun to study the differences in the flavor compounds in the Cabernet Sauvignon wines from these sub-regions. Zhang et al. [15] found that the Cabernet Sauvignon wines from the Eastern Foothills of Helan Mountain region were mainly characterized by higher concentrations of 1-decanol, isoamyl alcohol, isobutanol, benzyl alcohol, ethyl caprate, isoamyl acetate, citronellol, and *p*-cymene. Ge et al. [16] reported that the HSP, YN, and Yingchuan (YC) wines were well differentiated, while SZS and QTX wines had similar profiles of volatile compounds. However, these studies did not discuss the relationship between wine aroma characteristics and the terroir from different sub-regions of the Eastern Foothills of Helan Mountain in Ningxia, and these studies also have not been extended to sensory characteristics.

With the development of sensory science, check-all-that-apply (CATA), rate-all-that-apply (RATA), quantitative descriptive analysis (QDA), Sorting task, Rate-K attributes, and other methods have all been used in the studies of wine evaluation [17]. Among them, the CATA methodology is considered to be quick and simple and can be applied by trained or untrained assessors, making it easy to implement and useful in conjunction with classic methodologies to provide more accurate results [18]. For example, Rinaldi et al. [19,20] used CATA to evaluate the astringency sub-qualities of Sangiovese wines. Megan et al. [21] also used the CATA method to evaluate and characterize the aroma profiles of Syrah wines. In addition to chemical methods for the precise quantification of the aroma compounds of wines, there is also a need for sensory quantification. QDA is one of the methods for sensory quantification, and it has been widely used in the sensory evaluation of wines [22]. For example, previous studies showed that QDA was used to evaluate the intensity of aroma characteristics in Pinot Noir, Merlot, Hungarian, and Cabernet Franc wines [23,24,25].

In this study, sensory characteristics and volatile profiles of Cabernet Sauvignon wines from six sub-regions of the Eastern Foothills of Helan Mountain in Ningxia were investigated using CATA, QDA, and volatile chemical assay. One-way ANOVA, correspondence analysis (CA), and O2PLS-DA were carried out to identify the key features in Cabernet Sauvignon wines that drove the sub-regional differences in the Eastern Foothills of Helan Mountain. To our knowledge, this study is the first extensive regionality study using sensory evaluation (CATA and QDA) and chemical analysis on red wines from the different sub-regions of the Eastern Foothills of Helan Mountain in Ningxia, China.

## 2. Results and Discussion

### 2.1. Oenological Parameters of Wines from Different Sub-Regions

The ANOVA results of the oenological parameters, including alcohol, total acid, volatile acidity, total sugar, and pH, are listed in Table 1. Total acidity and total sugar were similar among the six sub-regions. However, significant differences in alcohol, volatile acidity, and pH were observed. HL wines had the highest level of alcohol, followed by SZS, YN, QTX, and HSP wines. The higher level of alcohol in the HL wines was probably due to the lower rainfall and longer sunshine duration in HL compared with the other sub-regions (Appendix A). A previous study also showed that wines from dry, warm regions had higher contents of alcohol than wines from wet, cool regions [15]. The SZS wines had the highest levels of volatile acidity, followed by the QTX wines. Beyond that, the results showed no further significant differences.

### 2.2. Volatile Profiles of Cabernet Sauvignon Wines from Different Sub-Regions

A total of 95 volatile compounds were quantified (Appendix A). O2PLS-DA was applied to identify the differences in the volatile profiles of Cabernet Sauvignon wines from six sub-regions in the Eastern Foothills of Helan Mountain. As shown in Figure 2, a clear separation was obtained by a reliable O2PLS-DA model (R^2^X = 0.743, R^2^Y = 0.652, Q^2^ = 0.448), except for the similarity between the XX and YN Cabernet Sauvignon wines. The XX and YN wines were characterized by higher concentrations of higher alcohols, C6 alcohols, esters (e.g., ethyl acetate, ethyl succinate, ethyl isobutanoate, ethyl 2-methylbutanoate, ethyl 3-methylbutanoate, and ethyl 2-hydroxy-4-methylpentanoate), and volatile phenols. In addition, the YN wines also had a high level of furans. As expected, XX and YN are adjacent to each other, and similar aromatic profiles were detected in their dry red wines. The previous studies had concluded that their similar aroma characteristics were due to their geographical proximity [9]. The similar volatile profiles of the two sub-regions XX and YN might be due to the fact that these two sub-regions are next to each other and influenced by their similar terrain and landforms, resulting in similar climates. As shown in Appendix A, XX and YN had similar average annual effective temperatures and sunshine durations. Both the HL and the QTX wines had personalities and were completely distinguished from the wines of the other regions. The HL wines had relatively high concentrations of esters (e.g., ethyl decanoate, ethyl dodecanoate, ethyl hexadecanoate, isoamyl acetate, ethyl nonanoate and isoamyl octanoate), fatty acids, lactones, and norisoprenoids, while the QTX wines had relatively high concentrations of C6 alcohols and esters (e.g., ethyl (*E*)-3-hexenoate, hexyl acetate, and methyl salicylate). The Cabernet Sauvignon dry red wines from SZS showed a noticeable difference, but their volatile profiles were similar with HSP wines. According to Figure 2b, the wines from SZS and HSP were mainly distinguished by their relatively higher concentrations of phenolic aldehydes and volatile phenols.

### 2.3. Sensory Characteristics of Cabernet Sauvignon Wines from Different Sub-Regions of Ningxia

CATA was used to characterize the aroma profiles of 71 Cabernet Sauvignon dry red wines from the Eastern Foothills of Helan Mountain. According to model optimization results, significant differences (*p* ≤ 0.05) were found with respect to 37 out of 49 attributes. CA was applied to visualize the relationships between 37 aroma attributes and 71 wines from six sub-regions (Figure 3). The results of CA showed a total explanation of 36.33% for F1 and F2. As shown in Figure 3, several HL and XX wines were characterized by green pepper, green, and mint in the first quadrant of CA. The second quadrant shows that approximately half of the wines from XX and YN were identified by the panelists as having prominent aromas such as cherry, mulberry, blackberry, raspberry, apple/pear, violet, sophora flower, strawberry, and jam. In the third quadrant, some of wines from HL, YN, and QTX were characterized by banana, vanilla, cream, baked sweet potatoes, caramel/taffy, chocolate, and toast. In addition, the fourth quadrant showed that some of the wines from the HL, XX, YN, and HSP sub-regions were also characterized by a high perception of aging aromas, such as spices, oak, toast, cream, vanilla, and coffee. One of the SZS wines was characterized by oak, coffee, and toast. However, only the aroma attributes of certain wines could be identified, so, in this study, typical wines from each sub-region were selected to analyze the different aroma characteristics of the different sub-regions more accurately. 

### 2.4. Typical Characteristics of Cabernet Sauvignon Wines in Different Sub-regions of the Eastern Foothills of Helan Mountain

Although the volatile profiles and sensory characteristics of the six sub-regions have been exhibited, the differences in the concentrations of aroma compounds among the sub-regions are unclear. As shown in Appendix A, the main aroma characteristics of wines from the Eastern Foothills of Helan Mountain were obtained. Those frequencies of attributes that were greater than 20% were summarized. To be more specific, both XX and YN wines were identified as having aromas of cherry, violet, mulberry, strawberry, hawthorn, raspberry, green pepper, green, jam, and spices. Besides, XX wines also had higher frequencies of aromas of sophora japonica and peach/apricot. Both HL and QTX wines were identified as having aromas of cherry, raspberry, green pepper, violet, and jam. HL wines were also identified as having strawberry, green, grass, and apple/pear attributes. QTX wines were also identified as having aromas of mulberry, green and spices. Both SZS and HSP wines were identified as having aromas of cherry and raspberry. SZS wines were also identified as having aromas of strawberry, hawthorn, raspberry, and jam. HSP wines were also identified as having aromas of mulberry, violet, spices, sophora japonica, and peach/apricot. The typical wines were selected based on the frequency results, and then the 27 representative wines in different sub-regions were determined by a bench tasting. To further clarify the influence of terroir on the wine aromas in different sub-regions, the chemical and sensory characteristics of the 27 typical wines were summarized and discussed.

#### 2.4.1. Volatile Compounds

There were 43 compounds in the selected representative wines that differed significantly among five sub-regions (*p* < 0.05) based on one-way ANOVA (Appendix A). The calculated aroma activity values (OAVs) of the aroma compounds in the 27 representative wines in are listed in Table 2 to evaluate the contribution of these compounds, and the volatile compounds with OAVs above 1.0 were regarded as contributing highly to the wine aroma [26,27]. To explain the differences in volatile compounds among the sub-regions and to explore the key aroma compounds in each sub-region, the marker volatile compounds with differences were concluded and discussed according to their origins and chemical categories. 

Methoxypyrazines originate mainly from grape berries and contribute to the green pepper and green notes in wines [28]. As shown in Table 2, only the concentration of IBMP was significantly different among the five sub-regions. The HL wines had the highest concentration of IBMP, followed by the QTX wines (Appendix A). Meanwhile, the IBMP concentration was found to be the highest in the HL-10 wine (16.73 ng/L). As discussed above, the high levels of alcohol in HL wines could be ascribed to the fact that sugar tends to accumulate quickly in the grapes in the warm and dry HL region. However, sugar, acidity, aroma compounds, and phenolic compounds are influenced by climatic conditions in different ways, which results in the inconsistency in the ripeness of these compounds. In this study, the high concentration of IBMP in the HL wines could be explained by the possibility that the degradation of IBMP was not adequate due to a short ripening period. Bogart and Bisson [29] reported that the concentration of IBMP could be ascribed to the effects of environmental factors (e.g., shade, moisture, and nitrogen fertilizer application) on its accumulation and to the degradation of the grape berries. Previous studies had also indicated that shading increased the IBMP concentration in Cabernet Franc berries, and both warm growing conditions and exposure conditions decreased IBMP in Cabernet Sauvignon berries [30,31]. The concentration of IBMP may be affected by the different climates of the vintage. For example, the grapes had a higher IBMP concentration throughout the ripening period in the humid vintage, although there were higher temperatures and more sunshine in September and October [32]. In addition, high humidity in the air before veraison with the growth of mold could increase the IBMP in wines [32,33]. Previous studies also showed that the IBMP concentration of Cabernet Sauvignon grapes in the GT and CL regions of China tended to decrease during the veraison period until harvest [34]. In this study, although QTX was very little affected by the barrier of Helan Mountain, the higher concentration of methoxypyrazines in QTX wines might be due to its lowest effective accumulated temperature compared to other sub-regions (Appendix A). This result was consistent with previous studies showing that grapes from cooler climates accumulated more methoxypyrazines during the growing period, resulting in wines with more prominent aromas of green plants (e. g., green, green pepper, plants) [8]. 

Terpenes and norisoprenoids originate mainly from grape berries and contribute to the floral and sweet notes in wines [35]. As shown in Table 2, 4-terpinenol showed a significant difference in five sub-regions, and its concentration was significantly higher in HSP wines (Appendix A). These phenomena could be associated with the higher sunshine duration from April to October in the HSP sub-region (Appendix A), which might induce the accumulation of terpenes. Previously, sunshine exposure was reported to increase the content of free-form and bound-form monoterpenes (e.g., *β*-citronellol, linalool, and geraniol) in Pinot Noir grape berries [36]. Among the norisoprenoids, only *β*-damascenone was significantly different among the five sub-regions and had relatively low olfactory thresholds and high OAVs (HL 58.60, XX 56.60, YN 56.00, QTX 61.40, HSP 37.00). The highest concentration was found in the QTX wines (Appendix A), followed by the HL, XX, YN, and HSP wines. The QTX sub-region has the lowest effective accumulated temperature and the highest sunshine duration from April to October among the five sub-regions (Appendix A), which could lead to a high concentration of total norisoprenoids. This result was consistent with a previous study which reported that the *β*-damascenone and total norisoprenoids were higher in Cabernet Sauvignon ripening grapes in cooler climates than in warmer climates [37]. However, Asproudi et al. [38] concluded that grapes grown in vineyards with higher temperatures and more sunshine had higher concentrations of norisoprenoids. In addition, numerous studies have focused on the influence of microclimatic conditions on norisoprenoid accumulation in grapes. For example, the concentration of *β*-damascenone in grape berries decreased because of short period weather fluctuations, such as the rainfall within a week [34]. Lee et al. [39] reported that *β*-damascenone was higher in leaf removal berries than in shading berries. However, other researchers found that exposure treatment did not affect the accumulation of *β*-damascenone in grapes [40]. Therefore, the mechanism of the influence of the terroir factors on damascenone accumulation is complex, and needs to be studied further.

Esters originate mainly from fermentation and contribute mainly to fruit notes in wines [41]. As shown in Table 2, the esters with significant differences among the five sub-regions and with an OAV more than 1.0 were ethyl octanoate, ethyl decanoate, ethyl butanoate, ethyl hexanoate, and isoamyl acetate. These compounds were also considered to be the most important odorants (OAV > 1.0) affecting the sensorial properties of Cabernet Sauvignon and Merlot wines from Ningxia, and were also responsible for the fruity, floral, and anise notes of the young wine [42]. The highest concentrations of these fruity esters were found in the HL wines, followed by XX wines (Appendix A). 

Higher alcohols and fatty acids are also the main sources of the compounds associated with fermentation and are produced through the metabolism of sugars or the corresponding amino acids (e.g., valine, leucine, isoleucine, and phenylalanine) [43,44]. As shown in Table 2, the higher alcohols with an OAV above 1.0 among the five sub-regions were isobutanol, isopentanol, and methionol. These compounds were most abundant in HL wines (Appendix A) and contributed to fusel and alcoholic aromas in wines [26]. The fatty acids with significant differences among the five sub-regions were acetic acid, hexanoic acid, and octanoic acid. Acetic acid as the main by-product of fermentation was found in the highest concentration in XX wines and in the lowest in HSP wines. Hexanoic acid was the highest in QTX wines, and octanoic acid was the highest in HL wines, which mainly contributed to the cheese and fat notes in the wines [45].

Phenolic aldehyde, volatile phenols, lactones, furans, and other compounds are mainly derived from oak barrels and influenced by the wine aging procedure [46]. Among phenolic aldehydes, vanillin was the highest in the HL wines (Appendix A) and mainly contributed to a vanilla note in the wine aroma [47]. The vanillin threshold is relatively low and plays an important role in wine sensory (OAV vanillin = 2.78). Lactones can originate from oak and also from fermentation [48]. In our study, the OAV of *γ*-decalactone was greater than 1.0, and it was highest in the HSP wines, contributing mainly to caramel, peach, and dried fruit aromas in the wines [49]. The OAVs of other compounds were below 1.0 and were regarded as less important in the regional differences in Cabernet Sauvignon wine aromas. However, phenolic aldehyde, volatile phenols, lactones, furans, and other compounds were less influenced by the *terroir*.

#### 2.4.2. Sensory Characteristics

To further clarify the aroma sensory characteristics of different sub-regions of Ningxia, the 27 representative wines were subjected to QDA (Appendix A). According to the results from Panel Check, all 10 categories of descriptors were significant (*p* < 0.0001), indicating that the 17 panelists could clearly identify the attributes of the 10 categories for all of the samples (Appendix A). The QDA results were subjected to the one-way ANOVA as shown in Table 3. Although the difference was not significant (*p* > 0.05), HL wines showed a relatively high intensity of herbaceous notes. This result was not only consistent with the results of the differences in the CATA results (Figure 3), but also with the quantitative result that showed a higher level of IBMP in the HL wines (Table 2). The intensity of black berry differed significantly among the five sub-regions, with the HSP wines showing significantly lower black berry intensity than the other four sub-regions. Red berry and fresh fruit were the same as black berry, with no significant differences between the sub-regions. However, both red berry and fresh fruit had the highest intensities in QTX wines, which may be explained by the higher concentrations of esters (e.g., ethyl octanoate, ethyl decanoate, ethyl butanoate, ethyl hexanoate, isoamyl acetate) in QTX wines (Appendix A). The highest intensity of floral in HSP wines may be due to their higher concentrations of norisoprenoids, which contributed to floral and sweet aroma in wines [50], followed by the QTX wines. The difference in spices was significant among five sub-regions, with the HSP wines showing the highest intensity. HSP wines also represented a high perception of smoke (*p* < 0.05), which was probably associated with the wineries’ winemaking style and barrel aging conditions [51]. Our study also found that herbaceous and black berry scores were the highest in HL. 

## 3. Materials and Methods

### 3.1. Wine Samples

A total of 71 commercial Cabernet Sauvignon dry red wines obtained from 43 wineries (approximately half of the wineries in Ningxia) of six sub-regions (including SZS, HL, XX, YN, QTX, and HSP sub-regions) in the Eastern Foothills of Helan Mountain in Ningxia were investigated (Appendix A). The vintages of most samples ranged from 2017 to 2019. Most of the wine samples were collected from the XX and YN sub-regions, which are relatively matured sub-regions, while fewer samples were collected from SZS sub-region because it is a developing wine-producing region with only two wineries able to provide monovarietal Cabernet Sauvignon wines. The distribution of the wineries on the map is shown in Figure 1. All of the commercial wines were made from pure *Vitis vinifera* L. Cabernet Sauvignon. The climatic conditions of the six sub-regions for the last three decades are shown in Appendix A. Details of the oenological parameters and the wine samples for the commercial wines from each sub-region are shown in Appendix A. The WineScan (FT 120) rapid-scanning infrared Fourier-transform spectrometer was used to measure the oenological parameters of the wines [10]. 

### 3.2. Reagents and Standard

Analytical grade reagents, including Milli-Q water, sodium hydroxide (NaOH), anhydrous sodium sulfate (Na_2_SO_4_), sodium chloride (NaCl), ammonium sulfate [(NH_4_)_2_SO_4_], tartaric acid, and glucose, were purchased from Beijing Chemical Works (Beijing, China) and Shanghai Macklin Biochemical (Shanghai, China). Chromatographic grade reagents, including methanol, ethanol, and dichloromethane, were bought from Fisher (Fairlawn, NJ, USA) and Honeywell (Marris Township, NJ, USA). Reference standards of volatile compounds and C_6_-C_24_ *n*-alkanes were purchased from Sigma-Aldrich (St. Louis, MO, USA). 

### 3.3. Analysis of Volatile Compounds

The extraction and quantitation of the methoxypyrazines of the wines were performed by HS-SPME-GC-MS according to our published method [52]. The samples of the wines were diluted to a 5.4% (*v*/*v*) alcohol with model solution (7 g/L tartaric acid and 2 g/L glucose without ethanol, pH 7.0), and the diluted samples were adjusted to pH 7.0 using NaOH solution. A 20 mL vial with 1.5 g NaCl was infused with 5 mL of diluted sample and was capped with a PTFE–silicon septum. After balancing at 38 °C for 10 min, the SPME fiber (2 cm DVB/CAR/PDMS 50/30 µm) was used to extract the target compounds in the headspace of the vial for 65 min. Then, the fiber was thermally desorbed at 250 °C for 8 min in splitless mode. The methoxypyrazines were detected by the Agilent 7890A GC–5975C MS (Agilent Technologies, Inc. Santa Clara, CA, USA). An lHP-INNOWAX capillary column (60 m × 0.25 mm id, 0.25 µm film thickness, J&W Scientific, Folsom, CA, USA) was used for separation. The flow rate of the carrier gas (helium) was set as 1 mL/min. The GC conditions: holding at 50 °C for 1 min, increasing to 110 °C with a rate of 3.0 °C/min, and increasing to 131 °C with a rate of 1.5 °C/min. The temperature of the after-run was set at 220 °C for 10 min. The MSD used the electron ionization (EI) mode at 70 eV and selected ion monitoring mode (SIM) for the detection of 3-isobutyl-2-methoxypyrazine (IBMP, *m*/*z* 124, 151, 166), 3-sec-butyl-2-methoxypyrazine (SBMP, *m*/*z* 124, 138, 151), and 3-isopropyl-2-methoxypyrazine (IPMP, *m*/*z* 137, 152). The quantification ions for IBMP, SBMP, and IPMP were *m*/*z* 124, 124, and 137, respectively. Each analysis was performed in duplicate. MPs were identified by comparing the retention times (RT) and the obtained mass spectra with those of aroma reference standards, and were quantified through the calibration curves prepared in a synthetic model wine solution (13% *v*/*v* ethanol, 7 g/L tartaric acid, 2 g/L glucose, and pH 3.5) of eight dilution levels.

The extraction and quantitation of higher alcohols, esters, fatty acids, terpenes, and nor isoprenoids were performed by HS-SPME-GC-MS according to our published method [53]. A 20 mL vial with 1.5 g of NaCl was infused with 5 mL of wine sample, and 10 μL of internal standard 4-methyl-2-pentanol (1.0 g/L) was added to the vial before it was capped with a PTFE–silicon septum. After balancing at 40 °C for 30 min, the SPME fiber (2 cm DVB/CAR/PDMS 50/30 µm) was used to extract target compounds in the headspace of the vial for 65 min. Then, the fiber was thermally desorbed at 250 °C in split mode with a ratio of 5:1. The heating procedure on an HP-INNOWAX capillary column (60 m × 0.25 mm id, 0.25 µm film thickness) was as follows: 50 °C for 1 min and 220 °C for 5 min at a rate of 3 °C/min. The MSD was operated in electron ionization (EI) mode at 70 eV, and the acquisition was performed in full ion scan mode (SCAN) with a mass scan range of *m*/*z* 29–350 u. Each analysis was performed in duplicate. By comparing the retention indices (RI) and the obtained mass spectra of volatile reference standards in the NIST 11 MS database using AMDIS, non-oak aroma compounds were identified. According to the ten attenuation levels in the synthetic model wine, the aroma standard’s calibration curves were obtained. The non-oak origin aroma compounds were quantified through calibration curves built with the ratio of the peak area of the target compound to the peak area of the corresponding internal standard (4-methyl-2-pentanol) against the concentration of each compound.

The extraction and quantitation of volatile phenols, phenolic aldehydes, lactones, and furans were performed by liquid-liquid extraction (LLE)-GC-MS according to the published methods, with slight modifications [42]. A 20 mL wine sample with the addition of 10 μL of mixed internal standard (4 g/L *γ*-caprolactone, 2 g/L 3,4-dimethylphenol, 4 g/L *o*-vanillin) and 5 g of (NH_4_)_2_SO_4_ were placed in a 50 mL centrifuge tube. Then, 5 mL of dichloromethane were added to the centrifuge tube, shaken for 5 min and centrifuged at 10,000 rpm for 10 min. The dichloromethane fraction was obtained after centrifugation. Then, the aqueous phase was extracted twice by the same extraction method. The dichloromethane fractions were collected and combined, dried with 1.5 g of anhydrous Na_2_SO_4_, and concentrated to 1 mL under a stream of nitrogen. Before GC-MS analysis, the final extract needed to be filtered by the 0.22 µm organic membrane. The extract of 1 μL was injected in splitless mode at 250 °C. The heating procedure on an HP-INNOWAX capillary column (60 m × 0.25 mm id, 0.25 µm film thickness) was as follows: 50 °C at 7 °C/min to 120 °C, and held for 5 min; 2 °C/min to 200 °C/min; and then 10 °C/min to 240 °C, and held for 20 min. The MSD was operated in electron ionization (EI) mode at 70 eV and acquisition was performed in full ion scan mode (SCAN) with a mass scan range of *m*/*z* 30–350 u. Each analysis of wine samples was performed in duplicate. By comparing the retention indices (RI) and the obtained mass spectra of volatile reference standards in NIST 11 MS database using AMDIS, oak volatile compounds were identified. According to the ten attenuation levels in the synthetic model wine, the aroma standard’s calibration curves were obtained. The oak derived aroma compounds were quantified through calibration curves built with the ratio of the peak area of the target compound to the peak area of the corresponding internal standard (*γ*-caprolactone for furans and lactones, 3,4-dimethylphenol for volatile phenols, *o*-vanillin for phenolic aldehydes) against the concentration of each compound.

### 3.4. Sensory Analysis

CATA and QDA were conducted to evaluate the sensory characteristics of wine samples. Sensory analyses were performed in a sensory laboratory, which had professional evaluation equipment (individual booths at a controlled room temperature of 20 °C and International Standards Organization wine-tasting glasses). To eliminate the system error, the wine samples were provided randomly. All participants were informed that performing these actions was completely voluntary, and informed consent was obtained from them in full accordance with the 1975 Declaration of Helsinki. Ethical approval for the involvement of human subjects in this study was granted by China Agricultural University Research Ethics Committee, reference number CAUHR-20220901. 

#### 3.4.1. CATA

All 71 wine samples were evaluated by a panel of 16 experienced experts (Panel 1, 10 males and 6 females, ranging in age 26–58). The participants included professional sommeliers, winemakers, and faculty members. Participants were asked to attend two sessions, and each session was split into two sub-sessions (about 50 min each), which were separated by an imposed pause of 20 min. Then, an attribute vocabulary was generated by Panel 1. In subsequent sessions, six experienced experts (Panel 2, 3 males and 3 females, ranging in age from 26 to 40) who had evaluated 71 wines consensually defined the appropriate attributes from the vocabulary after two discussions. The final list consisted of the 49 attributes in Appendix A.

CATA analysis was conducted by a panel of 40 panelists (Panel 3, 26 females and 14 males, ranging in age from 18 to 30) including students, staff, and faculties from the Center for Viticulture & Enology (CFVE). Each panelist who participated in the CATA evaluation was trained with references of the 49 attributes and Cabernet Sauvignon dry red wines for 1 h. Panel 3 was familiar with the attributes from training with the references and real wine samples. Panelists attended three sessions on three different days. Each session was split into four sub-sessions (about 25 min each) with an imposed pause of 10 min. Panelists were asked to select aroma descriptors from the 49 attributes to describe each wine. Correspondence analysis (CA) was conducted through the descriptors with significant differences (*p* < 0.05) by non-parametric tests based on Cochran’s Q test. Pre-processing of data was performed by Microsoft Excel (Microsoft, Washington, DC, USA), and CA was done by XLSTAT 2019 (Addinsoft, New York, NY, USA).

#### 3.4.2. QDA

According to the CATA results, typical wines with the sensory characteristics of each sub-region were selected and confirmed by a bench tasting, and were then referred to as the representative wines for subsequent QDA (Appendix A). In the bench tasting, the six experts (Panel 4, 3 females and 3 males, ranging in age 23–30) also generated 10 categories of aroma descriptors (herbaceous, black berry, red berry, fresh fruit, floral, spices, vanilla, oak, caramel, and smoke) for QDA, based on the 49 sensory attributes.

Panel 5 consisted of 17 panelists (12 females and 5 males, ranging in age 22–32) selected from among the students, faculty, and staff of the Center for Viticulture & Enology (CFVE), who had been fully trained before the experiment in QDA [53]. Panelists were required to evaluate the intensity of the ten aroma attributes described above on an 11-point scale (0 = very low intensity, 10 = strong intensity). Before each formal session, two monovarietal Cabernet Sauvignon dry red wines were used to form a uniform standard of scoring and a consensus was reached by Panel 5. The selected representative wine samples were divided into four sub-sessions for scoring, which lasted for a total of 2.5 h, with a 5–10 min break in each sub-session. Pre-processing of the data was performed by Microsoft Excel (Microsoft, USA), and one-way and two-way ANOVA of QDA results was done by IBM SPSS Statistics 24 and Panel Check (Nofima, Tromso, Norway).

### 3.5. Statistical Analysis

All data pre-processing was performed by Microsoft Excel (Microsoft, USA). One-way ANOVA (*p* < 0.05, Duncan’s multiple range test) for volatile compounds was done by IBM SPSS Statistics 24 (International Business Machines Corporation, Chicago, IL, USA), and O2PLS-DA analysis for volatile compounds was conducted by SIMCA (version 14.1 from Umetrics, Malmo, Sweden).

## 4. Conclusions

In this study, the volatile characteristics and sensory profiles of Cabernet Sauvignon dry red wines from six sub-regions of the Eastern Foothills of Helan Mountain in Ningxia were analyzed using sensory evaluation combined with the determination of volatile compounds. The results showed that the concentrations of volatile compounds differed among the wines from the six sub-regions. Among them, XX and YN had similar volatile profiles for their dry red wines due to their geographical proximity. The wines from the SZS and HSP sub-regions were mainly characterized by relatively higher phenolic aldehydes and volatile phenols. The concentration of IBMP and some esters, including ethyl octanoate, ethyl decanoate, ethyl butanoate, ethyl hexanoate, and isoamyl acetate were the highest in HL, while the concentration of *β*-damascenone was the highest in the QTX wines. The methoxypyrazines and norisoprenoids of Cabernet Sauvignon dry red wines in the HL and QTX sub-regions were mostly related to their climates (lowest effective accumulated temperature and the highest sunshine duration from April to October). As for sensory evaluation, the HL and QTX wines displayed higher intensities of herbaceous, black berry, and red berry notes compared to other sub-regions. Other aspects of the terroir effect on the accumulation of flavor compounds in Cabernet Sauvignon wines from different sub-regions of the Eastern Foothills of Helan Mountain in Ningxia, such as soil and management measures, merit further investigation.

## Figures and Tables

**Figure 1 molecules-27-08817-f001:**
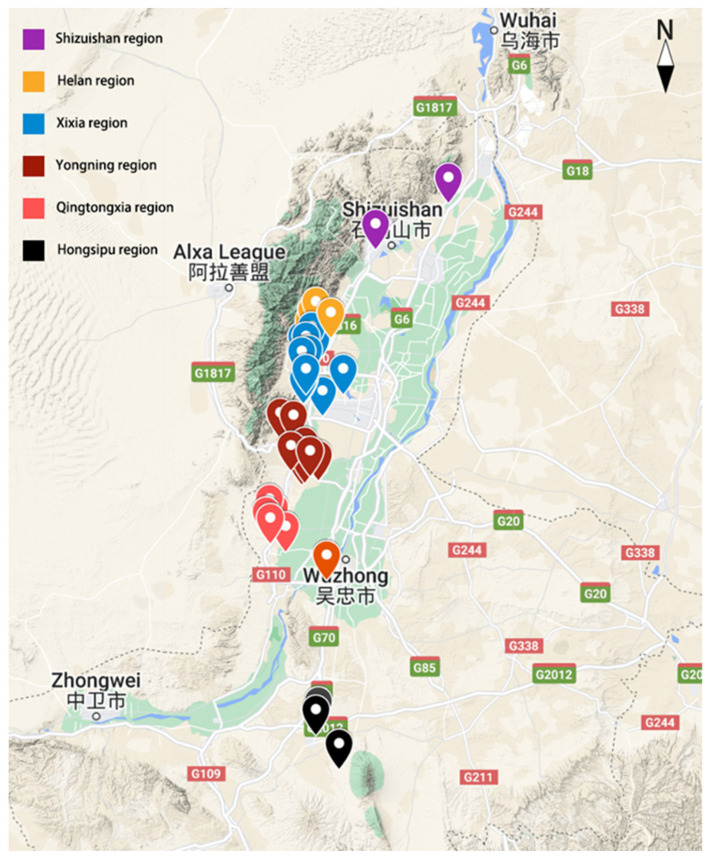
The terrain map of wineries in six sub-regions of the Eastern Foothills of Helan Mountain in Ningxia. The cities on the map are shown in English and Chinese.

**Figure 2 molecules-27-08817-f002:**
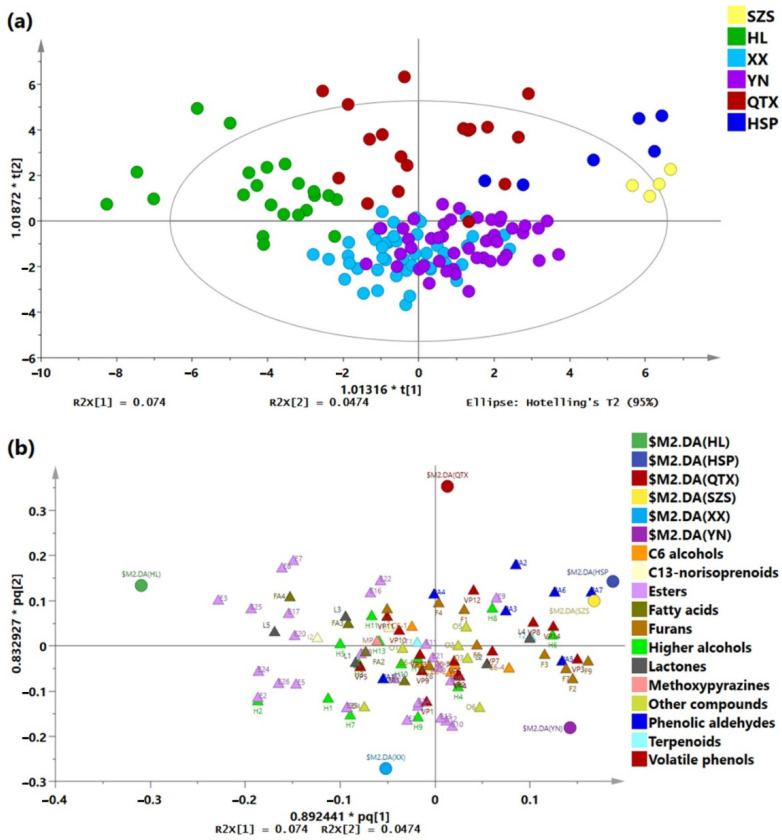
O2PLS-DA model based on the quantitative data of volatile compounds of Cabernet Sauvignon dry red wines from six sub-regions of the Eastern Foothills of Helan Mountain in Ningxia. (**a**) Score scatter plot for Cabernet Sauvignon dry red wines samples based on the replicate of each sample. (**b**) Loading plot for the volatile profiles of Cabernet Sauvignon dry red wines. The labels of volatile compounds are in accordance with Appendix A. SZS, Shizuishan region; HL, Helan region; XX, Xixia region; YN, Yongning region; QTX, Qingtongxia region; HSP, Hongsipu region.

**Figure 3 molecules-27-08817-f003:**
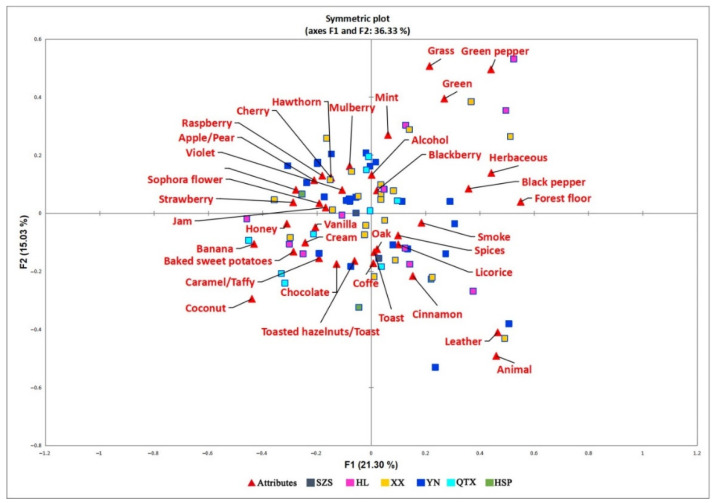
Correspondence analysis of ‘Cabernet Sauvignon’ dry red wines from six sub-regions of the Eastern Foothills of Helan Mountain in Ningxia. SZS, Shizuishan region; HL, Helan region; XX, Xixia region; YN, Yongning region; QTX, Qingtongxia region; HSP, Hongsipu region.

**Table 1 molecules-27-08817-t001:** Oenological parameters of Cabernet Sauvignon dry red wines from six sub-regions of the Eastern Foothills of Helan Mountain in Ningxia *.

Indexes	SZS	HL	XX	YN	QTX	HSP
Alcohol (%vol)	14.57 ± 0.12 ab	15.36 ± 0.75 a	14.81 ± 0.61 ab	14.47 ± 0.77 ab	14.43 ± 0.66 ab	14.17 ± 0.81 b
Total acidity (g/L)	5.60 ± 0.14 a	5.95 ± 0.6 a	5.71 ± 0.62 a	5.55 ± 0.43 a	5.60 ± 0.33 a	5.87 ± 0.67 a
Volatile acidity (g/L)	0.82 ± 0.08 a	0.63 ± 0.1 b	0.70 ± 0.11 ab	0.66 ± 0.1 b	0.67 ± 0.12 b	0.56 ± 0.01 b
Total sugar (g/L)	3.44 ± 0.52 a	4.28 ± 2.25 a	3.28 ± 0.78 a	3.27 ± 0.66 a	3.00 ± 0.78 a	4.23 ± 0.83 a
pH	3.93 ± 0.16 a	3.87 ± 0.12 ab	3.84 ± 0.13 ab	3.78 ± 0.17 ab	3.88 ± 0.15 a	3.68 ± 0.09 b

* The different lowercase letters represent significance at *p* < 0.05 among sub-regions, Duncan’s multiple range test. SZS, Shizuishan region; HL, Helan region; XX, Xixia region; YN, Yongning region; QTX, Qingtongxia region; HSP, Hongsipu region.

**Table 2 molecules-27-08817-t002:** OAVs of volatile compounds in Cabernet Sauvignon dry red wines from five sub-regions of the Eastern Foothills of Helan Mountain in Ningxia ^a^.

Compounds	Class ^b^	OAV				
		HL ^c^	XX	YN	QTX	HSP
Isobutanol	H	3.06	2.69	2.47	2.66	2.18
Isopentanol	H	14.45	13.54	12.39	12.35	10.74
1-Pentanol	H	<0.01	<0.01	<0.01	<0.01	<0.01
3-Methylpentanol	H	<0.01	<0.01	<0.01	<0.01	<0.01
2-Ethylhexanol	H	<0.01	<0.01	<0.01	<0.01	<0.01
2-Heptanol	H	- ^d^	-	-	-	-
2-Nonanol	H	-	-	-	-	-
1-Decanol	H	<0.01	<0.01	<0.01	<0.01	<0.01
Methionol	H	3.52	3.51	3.23	2.70	2.90
Benzyl alcohol	H	<0.01	<0.01	<0.01	<0.01	<0.01
Ethyl octanoate	E	107.84	71.67	64.85	47.21	48.79
Ethyl decanoate	E	1.90	1.06	1.06	1.44	0.61
Ethyl butanoate	E	8.89	7.86	7.05	7.17	6.26
Ethyl hexanoate	E	82.93	69.56	56.55	66.78	47.75
Ethyl (*E*)-3-hexenoate	E	-	-	-	-	-
Hexyl acetate	E	<0.01	<0.01	<0.01	<0.01	<0.01
Isoamyl acetate	E	25.27	16.17	13.56	14.40	15.13
Methyl salicylate	E	-	-	-	-	-
Isoamyl hexanoate	E	-	-	-	-	-
Isoamyl octanoate	E	0.04	0.04	0.03	0.04	0.03
Methyl octanoate	E	<0.01	<0.01	<0.01	<0.01	<0.01
4-Terpinenol	T	<0.01	<0.01	<0.01	<0.01	<0.01
*β*-Damascenone	I	58.60	56.60	56.00	61.40	37.00
3-Isobutyl-2-methoxypyrazine	MP	3.63	2.46	1.24	2.97	0.86
Acetic acid	FA	2.54	2.84	2.69	2.76	1.97
Hexanoic acid	FA	3.63	3.35	3.56	3.98	2.72
Octanoic acid	FA	2.66	2.12	2.22	2.89	1.74
*n*-Decanoic acid	FA	0.21	0.16	0.16	0.19	0.15
Vanillin	PA	2.78	1.42	2.16	1.59	0.44
Syringaldehyde	PA	-	-	-	-	-
Coniferaldehyde	PA	-	-	-	-	-
Acetosyringone	PA	-	-	-	-	-
Acetovanillone	PA	-	-	-	-	-
Phenol	VP	<0.01	<0.01	<0.01	<0.01	<0.01
4-Ethylphenol	VP	<0.01	<0.01	<0.01	<0.01	<0.01
4-Vinylguaiacol	VP	0.01	<0.01	0.02	0.05	0.03
Cresol	VP	-	-	-	-	-
*γ*-Decalactone	L	0.80	0.93	0.87	0.91	1.19
*γ*-Butyrolactone	L	0.32	0.20	0.18	0.20	0.17
Acetylfuran	F	-	-	-	-	-
2-Furanmethanol	F	-	-	-	-	-
Sotolon	F	0.32	0.54	0.56	0.58	0.69
(*E*)-2-Nonenal	O	-	-	-	-	-

^a^ The 43 volatile compounds with significant differences among the five sub-regions of the Eastern Foothills of Helan Mountain in Ningxia were summarized. ^b^ H, higher alcohols; E, esters; T, terpenoids; I, C13-norisoprenoids; MP, methoxypyrazines; FA, fatty acids; VP, volatile phenols; PA, phenolic aldehydes; L, lactones; F, furans; O, other compounds. ^c^ HL, Helan region; XX, Xixia region; YN, Yongning region; QTX, Qingtongxia region; HSP, Hongsipu region. ^d^ ‘-’ represents that the odor thresholds for these volatile compounds were not found.

**Table 3 molecules-27-08817-t003:** One-way ANOVA of QDA sensory evaluation of Cabernet Sauvignon dry red wines in sub-regions of the Eastern Foothills of Helan Mountain in Ningxia *.

Attributes	HL	XX	YN	QTX	HSP
Herbaceous	2.34 ± 0.36 a	1.70 ± 0.42 a	2.04 ± 0.66 a	2.01 ± 0.15 a	1.73 ± 0.04 a
Black berry	6.22 ± 0.15 a	5.93 ± 0.30 a	6.13 ± 0.20 a	6.14 ± 0.38 a	5.37 ± 0.77 b
Red berry	4.75 ± 0.24 a	4.53 ± 0.37 a	4.65 ± 0.26 a	4.76 ± 0.22 a	4.43 ± 0.40 a
Fresh fruit	2.59 ± 0.38 a	2.44 ± 0.65 a	2.66 ± 0.56 a	2.82 ± 0.30 a	2.29 ± 0.18 a
Floral	2.06 ± 0.25 b	2.04 ± 0.28 b	2.00 ± 0.38 b	2.31 ± 0.59 b	3.22 ± 1.09 a
Spices	1.89 ± 0.17 b	1.98 ± 0.29 ab	2.16 ± 0.42 ab	2.05 ± 0.19 ab	2.59 ± 1.00 a
Vanilla	1.91 ± 0.27 a	2.16 ± 0.46 a	2.20 ± 0.61 a	1.90 ± 0.21 a	1.69 ± 0.32 a
Oak	1.70 ± 0.37 a	2.05 ± 0.62 a	2.22 ± 0.74 a	1.85 ± 0.28 a	1.65 ± 0.27 a
Caramel	2.20 ± 0.16 a	2.75 ± 0.66 a	2.72 ± 0.83 a	2.54 ± 0.41 a	1.97 ± 0.35 a
Smoke	1.56 ± 0.36 b	1.53 ± 0.14 b	1.69 ± 0.37 b	1.79 ± 0.28 ab	2.27 ± 0.83 a

* The different lowercase letters represent significant at *p* < 0.05 among sub-regions, Duncan’s multiple range test. HL, Helan region; XX, Xixia region; YN, Yongning region; QTX, Qingtongxia region; HSP, Hongsipu region.

## Data Availability

Not applicable.

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
