# Peer review of "Volatile Profiles and Sensory Characteristics of Cabernet Sauvignon Dry Red Wines in the Sub-Regions of the Eastern Foothills of Ningxia Helan Mountain in China"

_molecules, 2022, doi:10.3390/molecules27248817_

Round 1
Reviewer 1 Report (New Reviewer)
This work is fine and moderately interesting for volatile profiles of Cabernet Sauvignon dry red wines in China. The results also can be believable based on the significant data presentation. However, before accepting for publication, small points of this work need to concern and a few modifications.
1. It seems like a bit elevated level of Plagiarism checking (authors should try to modify this vital point). The PDF file for checking is an attachment file herewith below, please check it out.
2. check the error typing in the whole manuscript.
3. In the conclusion section, the authors should try to summarize the significant point of the study or finding.

Author Response
Please see the attachment.

Reviewer 2 Report (New Reviewer)
Reviewer comments
In this study, the authors provided a wide range of evidence to explain the volatile profiles and the sensory characteristic of Cabernet Sauvignon wines from different sub-regions in China. The content of the manuscript is interesting. The methods and data visualization are worked well. The results are clearly stated and discussed. However, the manuscript needs revisions and the suggestions are revealed as follows.
1. In introduction, the Eastern Foothills of Helan Mountain in Ningxia has been fully displayed, but in introduced of the CATA and QDA methods (line 76-83), there is only one literature to describe the QDA methods. In the previous studies, there are a lot of literatures which used QDA to evaluate the wines. In my opinion, it should be added a few literatures.
2. Why do authors use the CATA instead of the RATA to evaluate the wines?
3. Why SZS wines are not present in Table 2? The zone abbreviation (line 432) is listed in the table caption.
4. There should be a space before and after “±” for the Table 1 and Table 3.
5. There are some grammars errors in the manuscript, please check throughout the manuscript and to ensure the grammars are correct.
Line 136 The MSD used the electron ionization (EI) mode at 70 eV and selecting ion monitoring mode (SIM) for should be “selected”
Line 249 Besides, the result also showed that total acidity and total sugar should be “ results”.
Line 274 HL wines is mainly relatively high in esters…should be “were”.
Line 306 …so we have selected typical wines from each sub-region… should be “had”.
Line361 Previous studies have also shown that the IBMP concentration of Cabernet Sauvignon grapes in the GT and CL regions of China…should be “had”.
Author Response
Please see the attachment.

This manuscript is a resubmission of an earlier submission. The following is a list of the peer review reports and author responses from that submission.
Round 1
Reviewer 1 Report
In this study, the authors studied the volatile profiles and the sensory characteristic of Cabernet Sauvignon dry red wines from different sub-regions in China. As stated in the abstract (Lines 13-16), the object of the present study was to "elucidate the different terroir on wine aroma in six-subregions...". Also, as stated in the introduction (Lines 33-35), "Terroir is the summary of regional climate, soil, geology, and management measures in the vineyard,..". Nevertheless, in this study many of these aspects were not considered, with the exception of some local metereological information, which, however refer to a time interval (from 1982 to 2011, see Table S2) which is far from the years of production of the analyzed samples (from 2015 to 2021, see Table S1). Also, the samples submitted to this study are not balanced in terms of number within the same year considered (see Table S1). In my opinion, it is quite difficult to obtain solid statistical information for these unbalanced sample groups and to obtain a possible correlation with the available climate information. For example, the authors only studied two samples of the year 2021 and, furthermore, for SZS the authors considered only two samples, one produced in 2015 and the other in 2019.
Furthermore, the authors do not describe exhaustively the approach used to annotate the volatile compounds listed in Table S5. Did they use reference compounds or information from a database? How did they explain the highly different quantitative ion values considered for some geometric isomers such as for example (E)-3-hexen-1-ol and (Z)-3-hexen-1-ol (see table S5)?
Besides, I have some concerns regarding the description of the chosen composition of the synthetic model wine solution (Lines 137-138) and how it was used during the study.
Overall, I found the research design not appropriate to justife the findings described in the conclusions.
Reviewer 2 Report
Dear Authors,
General opinion about the paper:
the paper has provided a wide range of evidence about Cabernet Sauvignon wines produced in China.
the methodology and data processing are well used
the results are clearly stated and explained
Specific opinion - minor changes to be done:
in introduction
literature 9 Gruner Veltliner should be erased
it is a white wine / grape variety
if You want to make a comparation you may take a substantial evidence - refernces about cabernet sauvignon wines worlwide, from UA, Australia, South Africa, Europe.
if possible after reference 11 state refernces for red wines characteristics of necesary
if not,
erase the sentences between ref 11 and next pasus (Nigxia Province.....)
Figure 2 should be enlarged and wider as Figure 3
because you cannot see the data points
best regards
the reviewer
Reviewer 3 Report
The study analyzed 71 Cabernet sauvignon wines from six growing zones, by terms of basic composition, volatile compounds, and sensory analysis. The data obtained contributed to the characterization of Cabernet sauvignon wines and on its linkage with sensorial data, but its use to discriminate growing zone may not be appropriate due to the lack of separation by vintages, as described below.
In my opinion a major issue is represented by the vintages considered. The authors used wines distributed from 2015 to 2021 vintages (Table S1), and analyzed wine results according to the growing zone, however they did not evaluate the effect of the vintage on wine composition. Vintage and subsequent wine storage can be very characterizing the wine aroma composition, especially for some categories such as fermentation esters (see: Waterhouse et al. 2016, doi:10.1002/9781118730720 chapters 7 and 25).
For instance, Helan wines (HL) were on average the youngest, given that more than 50% of them (6 wines on 11 total) were produced in the most recent 3 vintages. They also resulted in the highest OAV value for isoamyl acetate (Table 2) among the zones considered.
To exclude this factor from the evaluation of the differences among growing zones, in my opinion is necessary a separation by vintages of the data obtained.
Other additional factors involving the aroma evolution are the storage conditions (in cellar/in bottle, temperature, light,...) and the use of oak during the wine elaboration (we assume that young wines should have been not in contact with oak, while the opposite can be true for more aged wines). Data about the use of oak and the storage conditions before analysis is therefore necessary.
Other comments:
- I suggest to include Table S1 in the manuscript text and not in the supplementary material due to its importance in understanding the samples distribution.
- O2PLS-DA data: why there are four SZS points (and 6 HSP points) even if SZS wines were two and HSP wines considered were three (Table S1, Table S3)? Please clarify.
- Please provide the methods of analysis used for the evaluation of oenological parameters (e.g. Tables 1 and S3 compounds).
- Why SZS wines are not present in Table 2? The zone abbreviation is listed in the table caption.
Round 2
Reviewer 1 Report
In the revised manuscript, the authors justified the experimental design applied on the basis of the availability of vineyards in the sub-regions studied and the high costs required to obtain adequate meteorological data for the present study. However, I still have some concerns about the correctness of the experimental design and how the observed results were rationalized, as I discussed in my review.
In my opinion the manuscript is not suitable for publication on Foods.
Reviewer 3 Report
The authors replied to reviewers comments and performed some modifications to the proposed manuscript. However, the main point about vintages, also noted by another reviewer, is not resolved.
In the response to reviewers, the authors provided an elaboration based on wine vintages and wrote: "As shown in the Figure below, samples couldn’t be well distinguished by vintage". When looking at the provided figure (upper plot), to me it seems that vintage had an effect on the results obtained:
- the 2021 wine (orange; two replicates each wine) was located closely in the bottom left part of the plot
- all 2020 wines were located in the center left part of the plot comprised between x=-7 and x=-3
- except for three cases, all 2019 wines (light blue) were also distributed in the left part of the plot
- 2018 and 2017 wines were more sparse and variable
- 2016 wines (dark blue) were mostly located in the right side of the plot
- 2015 wines were all located in the right side of the plot
So, even if vintage may not be completely explaining the results, it seems clear that vintage affected these results, as expected given the degradation of certain aroma compounds during time (and possible new formation). To support the latter statement with the study data, isoamyl acetate (E17) position on the lower plot is on the extreme left part for the compounds analyzed (so the same side of young wines).
Given the information emerged from R1, in my opinion the authors need to re-think the data elaboration and the hypothesis, and in general the aims and scope of the study. For these reasons I think it is better to not continue with this submission procedure in order to provide additional time for preparation and eventual re-submission in this or other journals.